# Analysis of Gyro Bias Depending on the Position of Inertial Measurement Unit in Rotational Inertial Navigation Systems

**DOI:** 10.3390/s22218355

**Published:** 2022-10-31

**Authors:** Yeong-Bin Seo, Haesung Yu, Kyungdon Ryu, Inseop Lee, Juhyun Oh, Cheonjoong Kim, Sang Jeong Lee, Chansik Park

**Affiliations:** 1Weapon System Engineering, University of Science and Technology, Daejeon 34113, Korea; 2Agency for Defense Development, Daejeon 34186, Korea; 3Department of Electronics Engineering, Chungnam National University, 99 Daehak-ro, Yuseong-gu, Daejeon 34134, Korea; 4Department of Intelligent Systems and Robotics, Chungbuk National University, 1 Chungdae-ro, Seowon-gu, Cheongju 28644, Korea

**Keywords:** bias compensation, ring laser gyro (RLG), rotational inertial navigation system (RINS), system-level indirect calibration

## Abstract

In this paper, a calibration method for gyro bias that changes depending on the position of the IMU (inertial measurement unit) is proposed to improve the navigation performance of RLG-based RINS (ring-laser-gyro-based rotational inertial navigation system). RINS is a navigation device that compensates for the inertial sensor errors by utilizing the rotation of the IMU. In previous studies, the rotation scheme of the IMU is designed assuming that inertial sensor errors are not affected by position of the IMU. However, changes in temperature distribution, direction of gravity, and dithering according to the rotation of the IMU affect the inertial sensor errors, such as gyro bias. These errors could degrade the long-term navigation performance of RLG-based RINS. To deal with this problem, this paper proposed a compensation method of the gyro bias that changes depending on the position of the IMU. First, RINS is reviewed using a dual-axis 16-position rotation scheme and RLG. Next, the attitude error of RLG-based RINS is derived utilizing navigation equations. The effect of the gyro bias change caused by the change in the IMU attitude for the navigation performance of RINS is analyzed based on navigation equations and simulations. Finally, system-level indirect calibrations for the *Z*–axis up position and *Z*–axis down position are performed to calculate the gyro bias change caused by the IMU attitude. The accuracy of the proposed calibration method is verified by long-term navigation test. The test results show that the proposed calibration method improves the navigation performance of RINS compared with the conventional calibration method.

## 1. Introduction

INS (inertial navigation system) is a device that calculates navigation information such as position, velocity, and attitude using measurements of the IMU (inertial measurement unit) [1,2,3,4]. INS is widely used in various fields such as weapons systems, robots, and aerospace because it can calculate accurate navigation information without the help of external sensors. However, the navigation information calculated by INS has a problem that sensor errors such as bias, scale factor, and misalignment are accumulated over time. These sensor errors could degrade the long-term navigation performance of the INS.

To deal with this problem, a rotational inertial navigation system (RINS) is developed. RINS, a type of INS, operates in a way that the IMU is continuously rotated according to a given rotation scheme [5,6,7,8]. As RINS can suppress sensor errors through rotation of the IMU, it has the advantage of being able to improve navigation performance without the need for constructing IMU with a high-accuracy gyro or accelerometer. Because of these advantages, RINS is a widely used navigation device in areas such as marine navigation and submarines.

The navigation performance of RINS is mainly affected by gyro accuracy. Therefore, the structure of RINS is generally classified by gyro [5,6]. Previous studies developed RINS using gyros such as mechanical gyros, fiber optics gyros (FOGs), and ring laser gyros (RLGs) [7,8,9]. Among these gyros, RLG is the most widely used in RINS for marine navigation because RLG is less affected by acceleration errors and has a longer lifespan compared with other gyros. Therefore, this paper focus on the RLG-based RINS.

Since the development of RLG in 1908, various techniques have been developed to improve the accuracy of RINS designed based on RLG. The authors of [10] present a technique for improving the accuracy of azimuth estimation of RINS designed based on RLG through compensation of encoder output. The work of [11] proposes a technique for real-time RINS calibration using scale factor error of RLGs modeled on LS-SVMs. The authors of [12] studied the nonlinearity between the angular velocity measured during the operation of RLG-based RINS and the gyro scale factor error. The work of [13] presents a real-time calibration technique for the gyro scale factor error in RLG-based RINS. Furthermore, studies on advanced vibrational RLG are also being actively conducted [14,15].

Another factor to consider to improve the accuracy of RINS is the rotation scheme used to rotate the IMU. Gyro and accelerometer errors in the RINS operation phase can be directly compensated by rotating the IMU according to the precisely designed rotation scheme. Therefore, rotating the IMU by importing a precise rotation scheme is very important for RINS, which requires long-term navigation performance to be guaranteed [15].

The rotation scheme of RINS can be classified as a single-axis rotation scheme, dual-axis rotation scheme, and triple-axis rotation scheme based on the rotation axis of the IMU [16,17,18,19,20]. The single-axis rotation scheme rotates the IMU using a single-axis rotation table installed inside the RINS [21,22]. Compared with other rotation schemes, the single-axis rotation scheme can be implemented at a low cost, especially to precisely compensate for the horizontal axis error of RINS. However, single-axis rotation scheme could not compensate the vertical axis position error and vertical axis sensor errors. Therefore, a single-axis rotation scheme is not suitable as a rotation scheme of RINS designed for long-term navigation purposes.

Gyro and accelerometer errors in the RINS operation phase can theoretically be compensated by importing a rotation scheme of two or more axes [23,24]. RINS, designed by importing a triple-axis rotation scheme, has the advantage of compensating for accelerometer size effects and nonlinear errors in addition to sensor errors such as bias, scale factors, and misalignment [25,26]. However, a triple-axis rotation scheme-based RINS has the disadvantage of a complex structure and high implementation cost. In comparison, the dual-axis 16-position rotation scheme has the advantage of being able to compensate for gyro scale factor errors and non-alignment errors more precisely than the single-axis rotation scheme, as well as not having a higher system implementation cost than the triple-axis rotation scheme [27]. Because of these advantages, research on long-term navigation using RINS designed by importing a dual-axis rotation scheme is currently being actively conducted. The authors of [28] developed an eight-position rotation scheme that rotates the IMU using the dual-axis rotation table to compensate for the sensor error of the IMU. The authors of [29,30] presented techniques for rotating the IMU based on a 16-position rotation scheme and 32-position rotation scheme using the dual-axis rotation table.

The rotation schemes presented in the above study can effectively improve the theoretical navigation performance of RINS. However, in order to accurately compensate for sensor errors in the gyro and accelerometer by utilizing a rotation scheme, the sensor error should not be affected by the position of the IMU. If the sensor error changes depending on the position of the IMU rotating according to a given rotation scheme, the rotation scheme presented in the previous study cannot accurately compensate for the sensor error of RINS. In the case of RINS designed based on RLG, characteristics such as RLG internal temperature distribution, dithering, and direction of gravitational acceleration change depending on the rotation position of the IMU [31]. As the sensor error of RLG is affected by these characteristics, the sensor error of RLG also changes depending on the position of the IMU during the operation of RINS [32]. Therefore, in RLG-based RINS, there is a problem that the sensor error compensation accuracy of the rotation scheme suggested in previous research may be degraded [33,34,35].

To deal with this problem, this paper proposes a calibration method to improve the navigation performance of RLG-based RINS by pre-identifying and compensating for sensor errors in RLGs that change with the position of the IMU. The sensor error of the RLG affected by the position of the IMU was defined as a gyro bias according to the location and the system stage indirect calibration was introduced and identified.

In this paper, RINS is constructed using a three-axis RLG and a three-axis silicon accelerometer, and the rotation scheme of IMU is designed by importing a 16-position rotation scheme based on the dual-axis rotation table presented in [29]. The effect of gyro bias according to the position of the IMU on the navigation performance of RINS was analyzed through simulation. Finally, system-level indirect calibration was performed for various positions of the IMU to identify the gyro bias according to the position of the IMU. Navigation was performed using IMU set in the *Z*–axis up position and the *Z*–axis down position, and the gyro bias according to the position of the IMU was identified using the navigation information obtained during the navigation process. The gyro bias according to the location obtained through indirect calibration at the system stage was designed to be compensated in real time in the operation process of RINS. The navigation performance of the proposed calibration method was verified through long-term navigation test.

This paper is organized as follows. Section 2 describes the error model of RINS and the dual-axis 16 position rotation scheme for the IMU. Section 3 presents a procedure of identifying gyro bias that changes depending on the position of the IMU through system-level indirect calibration and conventional calibration methods. Section 4 verifies the performance of the proposed calibration method via long-term static state test. Finally, Section 5 presents the conclusion of this study.

## 2. Rotational Inertial Navigation System (RINS)

Coordinate frames applied in this paper are defined as shown in Table 1.

The body frame is applied to describe the position, velocity, and orientation of the sensor platform. The body frame consists of an origin that is typically placed at the center of gravity and three orthogonal axes. Generally, the axes in the body frame are configured to the body such that the *X*–axis is pointing forward, the *Y*–axis is pointing to the right, and the *Z*–axis is pointing up.

The earth frame is a global reference frame with its origin at the center of the Earth and three orthogonal axes fixed to the Earth. In the earth frame, the x, y, and z axes are aligned to local east, north, and up axes. We should point out that the earth frame rotates with the Earth.

The navigation frame is a frame that directs east–north–up (ENU). The navigation frame is fixed to the platform and moves with the body frame. The navigation frame contains three orthogonal axes in which the axis points to true east, the axis points towards the interior of the Earth, and the axis completes the right-handed system pointing north.

The orthogonal inertial frame has its origin at the center of the Earth. In contrast to the earth frame, the orthogonal inertial frame does not rotate with Earth.

### 2.1. Systematic Error Model of RINS

This section describes systematic errors that occur during the operation of RINS. The systematic error of RINS may be analyzed using the position and acceleration error model. The position error model of RINS may be expressed as follows using the error model of INS presented in [36].
(1)$${\dot{\phi }}^{n}=-\omega_{in}^{n}\times \phi^{n}+\delta \omega_{in}^{n}-\epsilon^{n},$$
where ϕ and $\dot{\phi }$ mean the attitude error and rates of the attitude error, respectively. $\omega_{in}^{n}$ is the earth rate and $\delta \omega_{in}^{n}$ is the earth rate error. $\epsilon^{n}$ is the gyro error of the navigation frame.

The acceleration error model of RINS may be expressed as follows in the same manner as the position error model.
(2)$$\delta {\dot{v}}^{n}=f^{n}\times \phi^{n}+\nabla^{n}-\left(2\omega_{ie}^{n}+\omega_{en}^{n}\right)\times \delta v^{n}-\left(2\delta \omega_{ie}^{n}+\delta \omega_{en}^{n}\right)\times v^{n}+\delta g^{n},$$
where δv and $\delta \dot{v}$ are velocity error and acceleration error for RINS, respectively. f means specific force and δg means gravitational error. $\nabla^{n}$ is accelerometer error for the navigation frame.

The gyro error in Equation (1) and accelerometer error in Equation (2) consist of static error such as white noise and random walk and deterministic error such as bias, scale factor, and misalignment [37]. Stochastic error cannot be modeled and has little effect on gyro error. The deterministic error occurs according to a certain tendency and has a greater effect on navigation performance than the static error. Therefore, the gyro error and the accelerometer error of the RINS are generally expressed using a deterministic error.

Let $\omega^{b}={\left[\omega_{x},\omega_{y},\omega_{z}\right]}^{T}$ be the input angular rate about body frame, the gyro error for navigation frame is defined as follows:(3)$$\begin{array}{ll} \epsilon^{n} & =C_{b}^{n}\cdot \delta \omega_{ib}^{b}\\ & =C_{b}^{n}\cdot \left(\left[\begin{array}{c} \beta_{x}\\ \beta_{y}\\ \beta_{z} \end{array}\right]+\left[\begin{array}{ccc} \beta_{xx} & \beta_{xy} & \beta_{xz}\\ \beta_{yx} & \beta_{yy} & \beta_{yz}\\ \beta_{zx} & \beta_{zy} & \beta_{zz} \end{array}\right]\cdot \omega^{b}\right), \end{array}$$
where $C_{b}^{n}$ is the coordinate transformation matrix for the body frame and $\delta \omega_{ib}^{b}$ is the gyro error for the body frame. $\beta_{x}$, $\beta_{y}$, $\beta_{z}$ are the gyro bias; $\beta_{xx}$, $\beta_{yy}$, $\beta_{zz}$ are the gyro scale factor error; and $\beta_{xy}$, $\beta_{xz}$, $\beta_{yx}$, $\beta_{yz}$, $\beta_{zx}$, $\beta_{zy}$ are the gyro misalignment.

Let $f^{b}={\left[f_{x},f_{y},f_{z}\right]}^{T}$ be the specific force about the body frame, the accelerometer error for the navigation frame is defined using the deterministic error of the accelerometer:(4)$$\begin{array}{ll} \nabla^{n} & =C_{b}^{n}\cdot \delta f^{b}\\ & =C_{b}^{n}\cdot \left(\left[\begin{array}{c} \alpha_{x}\\ \alpha_{y}\\ \alpha_{z} \end{array}\right]+\left[\begin{array}{ccc} \alpha_{xx} & \alpha_{xy} & \alpha_{xz}\\ \alpha_{yx} & \alpha_{yy} & \alpha_{yz}\\ \alpha_{zx} & \alpha_{zy} & \alpha_{zz} \end{array}\right]\cdot f^{b}\right), \end{array}$$
where $\delta f^{b}$ is the accelerometer error for the body frame. $\alpha_{x}$, $\alpha_{y}$, $\alpha_{z}$ are the accelerometer bias; $\alpha_{xx}$, $\alpha_{yy}$, $\alpha_{zz}$ are the accelerometer scale factor error; and $\alpha_{xy}$, $\alpha_{xz}$, $\alpha_{yx}$, $\alpha_{yz}$, $\alpha_{zx}$, $\alpha_{zy}$ are the accelerometer misalignment.

The gyro error and accelerometer error defined in Equations (3) and (4) are compensated through the rotation of the IMU during the operation of RINS. The navigation performance of RINS can be effectively improved by rotating the IMU according to an elaborately designed rotation scheme.

In this paper, the dual-axis 16-position rotation scheme presented in [29] is applied to rotate the IMU of RINS. The IMU used in RINS was designed using a three-axis RLG and a three-axis silicon accelerometer.

### 2.2. Navigation Performance of RINS Based on the Dual-Axis 16-Position Rotation Scheme

This section analyzes the navigation performance of RINS designed in Section 2.1 based on system error. As system error is mainly affected by gyro, the navigation performance of RINS can be determined using position error. The dual-axis 16-position rotation scheme imported to rotate the IMU of RINS in this paper consists of rotation sequences 1 to 4, rotation sequences 5 to 8, rotation sequences 9 to 12, and rotation sequences 13 to 16. The dual-axis 16-position rotation scheme is designed to perform four rotations at position A and return to the initial position A. In the process of performing the rotation scheme, the IMU rotates at a constant speed and stops at positions A, B, C, and D for a certain period of time. The rotation order of the dual-axis 16-position rotation scheme is shown in Figure 1.

In the dual-axis 16-position rotation scheme, the coordinate transformation matrices for position A, B, C, and D ${\left(C_{b}^{n}\right)}_{t}$(t=A,B,C,D) are shown in Equation (5).
(5)$${\left(C_{b}^{n}\right)}_{A}=\left[\begin{array}{ccc} 1 & 0 & 0\\ 0 & 1 & 0\\ 0 & 0 & 1 \end{array}\right],\;{\left(C_{b}^{n}\right)}_{B}=\left[\begin{array}{ccc} -1 & 0 & 0\\ 0 & -1 & 0\\ 0 & 0 & 1 \end{array}\right],\;{\left(C_{b}^{n}\right)}_{C}=\left[\begin{array}{ccc} -1 & 0 & 0\\ 0 & 1 & 0\\ 0 & 0 & -1 \end{array}\right],{\left(C_{b}^{n}\right)}_{D}=\left[\begin{array}{ccc} 1 & 0 & 0\\ 0 & -1 & 0\\ 0 & 0 & -1 \end{array}\right].$$

During the process of the rotation sequence, the angular rate of the IMU ${\left(\omega^{b}\right)}_{t}$
(t=1,…,16) for the body frame could be organized into Equation (6).
(6)$${\left(\omega^{b}\right)}_{1}={\left(\omega^{b}\right)}_{8}={\left(\omega^{b}\right)}_{11}={\left(\omega^{b}\right)}_{14}=\left[\begin{array}{c} 0\\ 0\\ \omega_{r} \end{array}\right],\;{\left(\omega^{b}\right)}_{2}={\left(\omega^{b}\right)}_{7}={\left(\omega^{b}\right)}_{12}={\left(\omega^{b}\right)}_{13}=\left[\begin{array}{c} \omega_{r}\\ 0\\ 0 \end{array}\right],\;{\left(\omega^{b}\right)}_{3}={\left(\omega^{b}\right)}_{6}={\left(\omega^{b}\right)}_{9}={\left(\omega^{b}\right)}_{16}=\left[\begin{array}{c} 0\\ 0\\ -\omega_{r} \end{array}\right],{\left(\omega^{b}\right)}_{4}={\left(\omega^{b}\right)}_{5}={\left(\omega^{b}\right)}_{10}={\left(\omega^{b}\right)}_{15}=\left[\begin{array}{c} -\omega_{r}\\ 0\\ 0 \end{array}\right],$$
where $\omega_{r}$ means the angular rates of the IMU in the process of conducting the rotation sequence.

For the dual-axis 16-position rotation scheme, the coordinate transformation matrices for rotation sequence ${\left(C_{b}^{n}\right)}_{t}$(t=1,…,16) are shown in Table 2.

The position error in the operation phase of RINS with dual-axis 16-position rotation can be calculated by integrating the rate of change in position error in Equation (1). As the time to perform dual-axis 16-position rotation is relatively short, this paper assumes that the effect of the Earth’s rotational angular velocity on the position error of RINS is negligibly small.

The position error occurring in the operation process of RINS is calculated as follows using the assumptions and Equations (1), (3), (5), and (6) and Table 2.
(7)$$\begin{array}{ll} \phi^{n} & =\overset{16T_{r}+16T_{s}}{\underset{0}{\int }}{\dot{\phi }}^{n}dt\\ & =\overset{16T_{r}+16T_{s}}{\underset{0}{\int }}\left(-\omega_{in}^{n}\times \phi^{n}+\delta \omega_{in}^{n}-\epsilon^{n}\right)dt\\ & \approx \overset{16T_{r}+16T_{s}}{\underset{0}{\int }}\epsilon^{n}dt\\ & =\overset{16T_{r}+16T_{s}}{\underset{0}{\int }}C_{b}^{n}\cdot \delta \omega_{ib}^{b}dt\\ & =\overset{T_{r}}{\underset{0}{\int }}\left({\left(C_{b}^{n}\right)}_{1}+{\left(C_{b}^{n}\right)}_{2}+{\left(C_{b}^{n}\right)}_{3}+{\left(C_{b}^{n}\right)}_{4}\right)\cdot \delta \omega_{ib}^{b}dt+\left({\left(C_{b}^{n}\right)}_{B}+{\left(C_{b}^{n}\right)}_{C}+{\left(C_{b}^{n}\right)}_{D}+{\left(C_{b}^{n}\right)}_{A}\right)\delta \omega_{ib}^{b}T_{s}\\ & \;\;+\overset{T_{r}}{\underset{0}{\int }}\left({\left(C_{b}^{n}\right)}_{5}+{\left(C_{b}^{n}\right)}_{6}+{\left(C_{b}^{n}\right)}_{7}+{\left(C_{b}^{n}\right)}_{8}\right)\cdot \delta \omega_{ib}^{b}dt+\left({\left(C_{b}^{n}\right)}_{D}+{\left(C_{b}^{n}\right)}_{C}+{\left(C_{b}^{n}\right)}_{B}+{\left(C_{b}^{n}\right)}_{A}\right)\delta \omega_{ib}^{b}T_{s}\\ & \;\;+\overset{T_{r}}{\underset{0}{\int }}\left({\left(C_{b}^{n}\right)}_{9}+{\left(C_{b}^{n}\right)}_{10}+{\left(C_{b}^{n}\right)}_{11}+{\left(C_{b}^{n}\right)}_{12}\right)\cdot \delta \omega_{ib}^{b}dt+\left({\left(C_{b}^{n}\right)}_{B}+{\left(C_{b}^{n}\right)}_{C}+{\left(C_{b}^{n}\right)}_{D}+{\left(C_{b}^{n}\right)}_{A}\right)\delta \omega_{ib}^{b}T_{s}\\ & \;\;+\overset{T_{r}}{\underset{0}{\int }}\left({\left(C_{b}^{n}\right)}_{13}+{\left(C_{b}^{n}\right)}_{14}+{\left(C_{b}^{n}\right)}_{15}+{\left(C_{b}^{n}\right)}_{16}\right)\cdot \delta \omega_{ib}^{b}dt+\left({\left(C_{b}^{n}\right)}_{D}+{\left(C_{b}^{n}\right)}_{C}+{\left(C_{b}^{n}\right)}_{B}+{\left(C_{b}^{n}\right)}_{A}\right)\delta \omega_{ib}^{b}T_{s} \end{array}$$

In Equation (7), $T_{r}$ is the rotation time of the IMU and $T_{s}$ is stationary time of the IMU.

In the process of conducting dual-axis 16-position rotation, the attitude error of RINS could be expressed as follows:(8)$$\phi^{n}=\overset{16T_{r}+16T_{s}}{\underset{0}{\int }}\epsilon^{n}dt=\overset{16T_{r}+16T_{s}}{\underset{0}{\int }}C_{b}^{n}\cdot \delta \omega_{ib}^{b}dt=\left[\begin{array}{c} 0\\ 0\\ 0 \end{array}\right].$$

As can be seen from Equation (8), gyro sensor errors such as bias, scale factor, and misalignment are compensated during the operation of RINS designed by applying dual-axis 16-position rotation. In the same way, accelerometer errors including bias, scale factor, and misalignment are compensated. Therefore, RINS in this paper can theoretically improve position errors on the east, north, and up axes.

## 3. Calibration Method for Gyro Bias Change Caused by Position of the IMU

### 3.1. Theorical Analysis of the Gyro Bias Change Caused by Position of the IMU

To improve the navigation performance of RINS by applying dual-axis 16-position rotation, the gyro sensor error should not be affected by the position of the IMU. However, the gyro error is affected by the gyro internal temperature distribution, gravitational direction, and dithering, which change with the position of the IMU. The gyro error affected by the position of the IMU can degrade the compensation accuracy of the gyro error such as bias, scale factor error, and non-alignment error, as well as the navigation performance of RINS. Therefore, in order to precisely compensate for the gyro error by performing dual-axis 16-position rotation, it is necessary to identify and compensate for the gyro error that varies with the location of the IMU.

In this section, the effect of the gyro error according to position of the IMU described in Section 2.2 is analyzed. This paper defined the gyro bias that changes according to the position of the IMU as $\delta \beta ={\left[\delta \beta_{x},\delta \beta_{y},\delta \beta_{z}\right]}^{T}$. The gyro bias that changes according to the position of the IMU is affected by the position of the IMU and the input angular velocity is not affected.

The gyro error of Equation (3) may be recalculated as follows by importing a gyro bias that changes according to the position of the IMU.
(9)$$\begin{array}{ll} \epsilon^{n} & =C_{b}^{n}\cdot \left(\delta \omega_{ib}^{b}+\delta \beta \right)\\ & =C_{b}^{n}\cdot \left(\left[\begin{array}{c} \beta_{x}\\ \beta_{y}\\ \beta_{z} \end{array}\right]+\left[\begin{array}{ccc} \beta_{xx} & \beta_{xy} & \beta_{xz}\\ \beta_{yx} & \beta_{yy} & \beta_{yz}\\ \beta_{zx} & \beta_{zy} & \beta_{zz} \end{array}\right]\cdot \omega^{b}+\left[\begin{array}{c} \delta \beta_{x}\\ \delta \beta_{y}\\ \delta \beta_{z} \end{array}\right]\right). \end{array}$$

The effect of the gyro bias, which changes according to the position of the IMU, on the position error of RINS is recalculated as follows using Equations (1), (5), and (6) and Table 2.
(10)$$\begin{array}{ll} \phi^{n} & =\overset{16T_{r}+16T_{s}}{\underset{0}{\int }}\epsilon^{n}dt\\ & =\overset{16T_{r}+16T_{s}}{\underset{0}{\int }}C_{b}^{n}\cdot \left(\delta \omega_{ib}^{b}+\delta \beta \right)dt\\ & =\overset{16T_{r}+16T_{s}}{\underset{0}{\int }}C_{b}^{n}\cdot \delta \omega_{ib}^{b}dt+\overset{16T_{r}+16T_{s}}{\underset{0}{\int }}C_{b}^{n}\cdot \delta \beta dt. \end{array}$$

According to Equation (10), the position error occurring during the operation of RINS is calculated by adding the integration of the gyro bias that changes according to the position of the IMU and the integration of the remaining gyro sensor error. As dual-axis 16-position rotation applied in this paper can compensate for the remaining gyro sensor error, the navigation performance of RINS can be improved by identifying and compensating for the gyro bias that changes according to the position of the IMU.

The position error occurring in the operation process of RINS is summarized as follows by substituting Equations (5) and (6) and Table 2 into Equation (10).
(11)$$\begin{array}{ll} \phi^{n} & =\overset{16T_{r}+16T_{s}}{\underset{0}{\int }}C_{b}^{n}\cdot \delta \beta dt=\overset{16T_{r}+16T_{s}}{\underset{0}{\int }}C_{b}^{n}\cdot \left[\begin{array}{c} \delta \beta_{x}\\ \delta \beta_{y}\\ \delta \beta_{z} \end{array}\right]dt\\ & =\overset{T_{r}}{\underset{0}{\int }}\left({\left(C_{b}^{n}\right)}_{1}+{\left(C_{b}^{n}\right)}_{2}+{\left(C_{b}^{n}\right)}_{3}+{\left(C_{b}^{n}\right)}_{4}\right)\cdot \delta \beta dt+\left({\left(C_{b}^{n}\right)}_{B}+{\left(C_{b}^{n}\right)}_{C}+{\left(C_{b}^{n}\right)}_{D}+{\left(C_{b}^{n}\right)}_{A}\right)\cdot T_{s}\delta \beta \\ & \;\;+\overset{T_{r}}{\underset{0}{\int }}\left({\left(C_{b}^{n}\right)}_{5}+{\left(C_{b}^{n}\right)}_{6}+{\left(C_{b}^{n}\right)}_{7}+{\left(C_{b}^{n}\right)}_{8}\right)\cdot \delta \beta dt+\left({\left(C_{b}^{n}\right)}_{D}+{\left(C_{b}^{n}\right)}_{C}+{\left(C_{b}^{n}\right)}_{B}+{\left(C_{b}^{n}\right)}_{A}\right)\cdot T_{s}\delta \beta \\ & \;\;+\overset{T_{r}}{\underset{0}{\int }}\left({\left(C_{b}^{n}\right)}_{9}+{\left(C_{b}^{n}\right)}_{10}+{\left(C_{b}^{n}\right)}_{11}+{\left(C_{b}^{n}\right)}_{12}\right)\cdot \delta \beta dt+\left({\left(C_{b}^{n}\right)}_{B}+{\left(C_{b}^{n}\right)}_{C}+{\left(C_{b}^{n}\right)}_{D}+{\left(C_{b}^{n}\right)}_{A}\right)\cdot T_{s}\delta \beta \\ & \;\;+\overset{T_{r}}{\underset{0}{\int }}\left({\left(C_{b}^{n}\right)}_{13}+{\left(C_{b}^{n}\right)}_{14}+{\left(C_{b}^{n}\right)}_{15}+{\left(C_{b}^{n}\right)}_{16}\right)\cdot \delta \beta dt+\left({\left(C_{b}^{n}\right)}_{D}+{\left(C_{b}^{n}\right)}_{C}+{\left(C_{b}^{n}\right)}_{B}+{\left(C_{b}^{n}\right)}_{A}\right)\cdot T_{s}\delta \beta \\ & =\left[\begin{array}{c} 0\\ 0\\ -8T_{s}\delta \beta_{z} \end{array}\right] \end{array}$$

As shown in Equation (11), the gyro bias that changes according to the position of the IMU during the RINS operation phase is compensated in the x–axis and y–axis. However, the z–axis gyro bias $\delta \beta_{z}$, which varies with the position of the IMU, causes an up-axis position error proportional to $T_{s}$ and $\delta \beta_{z}$. In the case of gyro errors including scale factor and misalignment and accelerometer errors including bias, scale factor, and misalignment, these errors are compensated for through the dual-axis 16-position rotation scheme.

For example, if $T_{s}$ is 180 s and $\delta \beta_{z}$ is 0.005 deg/h, the up-axis attitude error during the operation of RINS is approximately 0.002 deg. Because the up-axis attitude error leads to the latitude error and longitude error, it is necessary to compensate for the gyro bias that changes according to the position of the IMU to improve navigation performance.

In this paper, numerical simulation is performed to evaluate the effect of the gyro bias that changes according to the position of the IMU. The changes in navigation performance were compared by performing navigation simulations on conditions that compensate $\delta \beta_{x}$, $\delta \beta_{y}$, and $\delta \beta_{z}$ and non-compensating conditions. The position where the navigation simulation is performed is 36.6° latitude and 123.6° longitude and the navigation time is 1 week. The angular velocity $\omega_{s}$ of the IMU rotating according to the dual-axis 16-position rotation is 2 deg/s and the time $T_{s}$ of the IMU stopping at positions A, B, C, and D is 180 s. The sensor specification of RINS used in the simulation is shown in Table 3.

The position error obtained by performing a navigation simulation according to the conditions in Table 3 is shown in Figure 2.

The navigation simulation results show that the up-axis average attitude error when not compensating for the gyro bias that changes according to position of the IMU is 0.0338 deg and the up-axis average attitude error when compensating for the gyro bias that changes according to position of the IMU is 0.0302 deg. Consequently, the up-axis attitude error can be improved by 10.65% by compensating for the gyro bias that changes according to the position of the IMU.

The latitude error and longitude error obtained by performing the navigation simulation are shown in Figure 3 and the horizontal axis position error at the end of the navigation is summarized in Table 4.

As shown in Table 4, by compensating for the gyro bias that changes according to position of the IMU, the latitude error was improved by 0.53% and the longitude error was improved by 59.44%. Therefore, the position error of RINS could be effectively improved by compensating for the gyro bias that changes according to position of the IMU.

### 3.2. Identifyning Gyro Bias Change Caused by Position of the IMU

This section describes techniques for identifying and compensating for gyro biases that vary with the location of the IMU. In this paper, a system-level indirect calibration technique was adopted to identify the gyro bias that changes according to the position of the IMU and compensated according to the IMU position condition.

In system-level indirect calibration techniques, RINS performs navigation for a short period of time in a static state [33]. Under static state and short-term navigation conditions, it can be assumed that the effect of the transport rate and Coriolis effects on the navigation performance of RINS is negligibly small. Based on the above assumption, the position error of RINS in the process of performing the system-level indirect calibration technique is obtained again as follows using Equation (1).
(12)$$\begin{array}{ll} {\dot{\phi }}^{n} & =\left[\begin{array}{c} {\dot{\phi }}_{E}\\ {\dot{\phi }}_{N}\\ {\dot{\phi }}_{U} \end{array}\right]\\ & =-\omega_{in}^{n}\times \phi^{n}-C_{b}^{n}\cdot \left(\delta \omega_{ib}^{b}\right)\\ & =-\left[\begin{array}{ccc} 0 & +\phi_{U} & -\phi_{N}\\ -\phi_{U} & 0 & +\phi_{E}\\ +\phi_{N} & -\phi_{E} & 0 \end{array}\right]\left[\begin{array}{c} \Omega_{E}\\ \Omega_{N}\\ \Omega_{U} \end{array}\right]-C_{b}^{n}\cdot \left(\delta \omega_{ib}^{b}\right). \end{array}$$

By assuming that the effect of the earth rate on the position error of RINS is very small and residual gyro/accelerometer errors except gyro bias are sufficiently small, the position error of Equation (12) is summarized as shown in Equation (13).
(13)$${\dot{\phi }}^{n}\approx -C_{b}^{n}\cdot \beta =-C_{b}^{n}\cdot \left[\begin{array}{c} \delta \beta_{x}\\ \delta \beta_{y}\\ \delta \beta_{z} \end{array}\right].$$

In a similar manner, the acceleration error of RINS in the process of performing the system-level indirect calibration technique is obtained again as follows:(14)$$\begin{array}{ll} \delta {\dot{v}}^{n} & =\left[\begin{array}{c} \delta {\dot{V}}_{E}\\ \delta {\dot{V}}_{N}\\ \delta {\dot{V}}_{U} \end{array}\right]\\ & =f^{n}\times \phi^{n}+C_{b}^{n}f^{b}\\ & =\left[\begin{array}{ccc} 0 & +\phi_{U} & -\phi_{N}\\ -\phi_{U} & 0 & +\phi_{E}\\ +\phi_{N} & -\phi_{E} & 0 \end{array}\right]\left[\begin{array}{c} 0\\ 0\\ g \end{array}\right]+C_{b}^{n}\cdot \delta f^{b}. \end{array}$$

By assuming that the acceleration error of RINS does not change over time, the first derivative of the acceleration error over time in Equation (14) could be calculated as shown in Equation (15).
(15)$$\begin{array}{ll} \delta {\ddot{v}}^{n} & =\left[\begin{array}{c} \delta {\ddot{V}}_{E}\\ \delta {\ddot{V}}_{N}\\ \delta {\ddot{V}}_{U} \end{array}\right]\\ & \approx f^{n}\times {\dot{\phi }}^{n}\\ & =\left[\begin{array}{ccc} 0 & +{\dot{\phi }}_{U} & -{\dot{\phi }}_{N}\\ -{\dot{\phi }}_{U} & 0 & +{\dot{\phi }}_{E}\\ +{\dot{\phi }}_{N} & -{\dot{\phi }}_{E} & 0 \end{array}\right]\cdot \left[\begin{array}{c} 0\\ 0\\ g \end{array}\right]. \end{array}$$

The gyro bias that changes according to the position of the IMU may be calculated using the up-axis position error of Equation (13) and the horizontal axis acceleration error change rate of Equation (15). In this paper, system-level indirect calibration was repeatedly performed in the *Z*–axis up position and *Z*–axis down position to obtain the gyro bias that changes according to the position of the IMU.

The rotation sequence for calculating the gyro bias at the *Z*–axis up position consists of self-alignment at the initial position, navigation at the initial position, and navigation at the post-rotation position. The navigation time for the initial position and the post-rotation position is 10 min and the position of RINS is shown in Figure 4.

In this paper, analysis of the rotation sequence applies sum frames including the body frame and navigation frame. The detailed descriptions of these frames are presented in Section 2. In Figure 4, the azimuth angle of the initial position is 0 degrees and the azimuth angle of the post-rotation position is 180 degrees. The coordinate transformation matrix for the initial position and the post-rotation position is as shown in Equation (16).
(16)$${\left(C_{b}^{n}\right)}_{Z_{u}1}=\left[\begin{array}{ccc} 1 & 0 & 0\\ 0 & 1 & 0\\ 0 & 0 & 1 \end{array}\right],\;\;{\left(C_{b}^{n}\right)}_{Z_{u}2}=\left[\begin{array}{ccc} -1 & 0 & 0\\ 0 & -1 & 0\\ 0 & 0 & 1 \end{array}\right].$$

The gyro bias for the *Z*–axis up position is obtained by substituting Equation 16 into Equations (13) and (15).
(17)$$\begin{array}{c} \beta_{x,u}=\frac{1}{2g}\left({\left(\delta {\ddot{V}}_{N}\right)}_{z_{u}2}-{\left(\delta {\ddot{V}}_{N}\right)}_{z_{u}1}\right)\\ \beta_{y,u}=-\frac{1}{2g}\left({\left(\delta {\ddot{V}}_{E}\right)}_{z_{u}2}-{\left(\delta {\ddot{V}}_{E}\right)}_{z_{u}1}\right)\\ \beta_{z,u}=-\frac{1}{2}\left({\left({\dot{\phi }}_{U}\right)}_{z_{u}2}+{\left({\dot{\phi }}_{U}\right)}_{z_{u}1}\right) \end{array}.$$

$\beta_{x,u}$ could be determined using the second derivative of the north-axis velocity error and $\beta_{y,u}$ could be determined using the second derivative of the east-axis velocity error. $\beta_{z,u}$ could be calculated using the rates of up-axis attitude error before and after rotation.

The gyro bias for the *Z*−axis down position may be obtained in the same manner as the gyro bias for the *Z*−axis up position. The rotation sequence for obtaining the gyro bias at the *Z*–axis down position is shown in Figure 5.

In Figure 5, the coordinate transformation matrix for the initial position and the post-rotation position is as shown in Equation (18).
(18)$${\left(C_{b}^{n}\right)}_{Z_{d}1}=\left[\begin{array}{ccc} -1 & 0 & 0\\ 0 & 1 & 0\\ 0 & 0 & -1 \end{array}\right],\;{\left(C_{b}^{n}\right)}_{Z_{d}2}=\left[\begin{array}{ccc} 1 & 0 & 0\\ 0 & -1 & 0\\ 0 & 0 & -1 \end{array}\right].$$

The gyro bias for the *Z*–axis down position is obtained by substituting Equation (18) into Equations (13) and (15).
(19)$$\begin{array}{c} \beta_{x,d}=-\frac{1}{2g}\left({\left(\delta {\ddot{V}}_{N}\right)}_{z_{d}2}-{\left(\delta {\ddot{V}}_{N}\right)}_{z_{d}1}\right)\\ \beta_{y,d}=-\frac{1}{2g}\left({\left(\delta {\ddot{V}}_{E}\right)}_{z_{d}2}-{\left(\delta {\ddot{V}}_{E}\right)}_{z_{d}1}\right)\\ \beta_{z,d}=\frac{1}{2}\left({\left({\dot{\phi }}_{U}\right)}_{z_{d}2}+{\left({\dot{\phi }}_{U}\right)}_{z_{d}1}\right) \end{array}.$$

The gyro bias that changes according to the position of the IMU may be obtained using Equations (17) and (19). The gyro bias that changes according to the position of the IMU is repeatedly identified for various temperature conditions and positional conditions of the IMU.

The gyro bias that changes according to the position of the IMU obtained by performing the above process is compensated in real time in the operation process of the RINS. The process of realizing the proposed calibration method and verification tests are described in Section 4.

## 4. Experiment Results

In this section, the accuracy of the proposed calibration method in Section 3 is verified by performing stationary state tests. A 1 Nm/h class of RINS consisting of a three-axis RLG and a three-axis silicon accelerometer is used for the stationary state test. This paper constructed the IMU by adopting the sensor specification in Table 3. The RLG arrangement in the IMU for the *Z*-axis up position and *Z*-axis down position is described in Figure 6. During the operation of RINS, the IMU rotates according to the dual-axis 16-position rotation scheme presented in [29]. Gyro bias, which varies with the location of the IMU, was identified by performing system-level indirect calibration under various temperature conditions.

### 4.1. Experiment Setup

This section verifies the performance of the calibration technique proposed in this paper by performing stationary state tests. A 1 Nm/h class of RINS consisting of a three-axis RLG and a three-axis silicon accelerometer is used for the stationary state test. During the operation of RINS, the IMU rotates according to the dual-axis 16-position rotation scheme presented in [29]. Gyro bias, which varies with the location of the IMU, was identified by performing system-level indirect calibration under various temperature conditions.

The system-level indirect calibration performed to identify the gyro bias depending on the position of IMU is performed in the same order as shown in Figure 6 and Figure 7. The data processing PC is responsible for collecting navigation data and calculating calibration coefficients. The DC power supply supplies the power required for the system-level indirect calibration process. RINS is installed inside the temperature chamber of the rate table and rotates according to the input of the rate table control console.

The system step indirect calibration test is performed on the *Z*–axis up position condition of the IMU and the *Z*–axis down position condition. The calibration test is repeated 37 times in a temperature range between 6.54 °C and 58.14 °C. The rotational angular velocity of the IMU is 2 deg/s. During the process of the calibration test, navigation was conducted for about 1500 s.

Gyro bias depending on the position of IMU is calculated using Equations (17) to (19). To reflect the temperature effect on the gyro bias depending on the position of the IMU, the calculation process is repeated for various temperature conditions. Figure 8 shows that the gyro bias changes according to the position of the IMU identified by performing the calibration test.

As shown in Figure 8, in the *X*–axis and *Z*–axis, the effect of the gyro bias according to the position of the IMU is large. On the other hand, in the *Y*–axis, the effect of the gyro bias according to the position of the IMU is hardly shown.

These identified gyro bias sets are modeled as compensation functions based on third-order polynomial. The compensation functions are utilized for real-time compensation. Therefore, navigation performance of the RINS can be improved using an appropriate compensation function depending on the position of the IMU.

For RINS used in this paper, the *X*–axis gyro and the Y–axis gyro are arranged similarly in the *Z*–axis up position and *Z*–axis down position. However, the identified gyro bias changed by 0.015 deg/h on the *X*–axis. In the case of the *Z*–axis gyro, heat transfer by gravity from the electric board to the mono block can be expected to have an effect, but only 0.004 deg/h has changed. This difference in gyro bias appears differently depending on each RINS. Therefore, it can be interpreted that the calibration method proposed in this paper can improve the navigation performance of RINS.

The gyro bias identified under the *Z*–axis up position condition and the *Z*–axis down position condition varies by about 0.015 deg/h depending on the position of the IMU on the *X*–axis and by about 0.004 deg/h depending on the position of the IMU on the *Z*–axis. As analyzed in Section 3.1, the gyro bias that changes according to the position of the IMU may degrade the navigation performance of RINS. Therefore, it is possible to improve navigation performance by compensating for the gyro bias that changes depending on the position of the IMU identified through calibration tests during the operation of the RINS.

### 4.2. Results of the Static State Navigation Test

In this section, we validate the improved navigation performance by compensating for the gyro bias that changes according to the position of the IMU. In this paper, long-term stationary navigation was performed to determine the navigation performance of RINS. Long-term stationary navigation tests are designed to store the gyro bias obtained in Section 4.1 on the navigation computer of RINS, compensate for the gyro bias that changes according to the position of the IMU in real time, and perform the navigation. For the static-state navigation test, RINS is mounted on a leveled surface and performs the navigation for about 7 days. In this paper, the navigation performances of the proposed calibration method and conventional calibration method were evaluated based on the horizontal axis position error.

The results of long stationary navigation tests are used to assess the effect of gyro biases depending on the position of the IMU on navigation performance. In this paper, navigation performance was determined using the latitude error, hardness error, position error, and time root mean square (tRMS) position error at the end of the stationary navigation test [37]. The tRMS position error is calculated as in Equation (20).
(20)$$\mathrm{tRMS}\left(t_{n}\right)=\sqrt{\frac{{\left(E_{1}\right)}^{2}+{\left(E_{2}\right)}^{2}+\ldots +{\left(E_{n}\right)}^{2}}{n}}=\sqrt{\frac{1}{n}\cdot \overset{n}{\underset{i=1}{\sum }}{\left(E_{i}\right)}^{2}}.$$
where $t_{n}$ means navigation time and $E_{i}$ means the horizontal-axis position error for $t_{i}$.

Latitude error and longitude error obtained by performing the stationary state test for a long time are shown in Figure 9 and horizontal axis position error and tRMS error are shown in Figure 10. The navigation performance obtained by the stationary navigation test is summarized in Table 5.

The long-term stationary state test results show that the calibration technique proposed in this paper improves the navigation performance of RINS. Based on the navigation performance at the end of the long-term stationary state test, the tRMS position error of 1.2921 Nm was measured by applying the calibration technique proposed in this paper. On the other hand, when the gyro bias that changes according to the position of the IMU was not compensated for, a tRMS position error of 2.4603 Nm was measured.

On average, the navigation performance was improved by 69.83% by identifying and compensating for the gyro bias, which changes according to the position of the IMU. Applying the calibration technique proposed in this paper, the horizontal-axis position error was improved by 79.86% and the tRMS position error was improved by 47.48%. In particular, it was confirmed that the longitude error was improved by 82.12% through compensation for the gyro bias, which changes according to the position error of the IMU.

## 5. Conclusions

In this paper, to improve the navigation performance of RINS, we propose a calibration technique that identifies and compensates for the gyro bias that changes according to the position of the IMU. First, RINS was designed using RLG and IMU was rotated by importing the dual-axis 16-position rotation scheme. The effect of the gyro bias that changes according to the position of the IMU on the navigation performance of RINS designed in this paper was theoretically analyzed and verified through navigation simulation.

Gyro bias, which changes according to the location of the IMU, was identified by applying a system-level indirect calibration technique. By identifying the gyro bias for the *Z*–axis up position condition and the *Z*–axis down position condition of the IMU, the gyro bias that changes according to the position of the IMU was obtained. The gyro bias that changes according to the position of the IMU identified by repeating system-level indirect calibration under various temperature conditions is compensated for in real time during the operation of RINS.

The performance of the proposed technique was verified by performing long-term stationary navigation tests. A long-term stationary navigation test was performed according to the compensation condition of the gyro bias that changes depending on the position of the IMU, and the longitude error, latitude error, horizontal-axis position error, and tRMS position error were measured. As a result of the test, it was confirmed that the navigation performance of RINS was improved by up to 82.12% by applying the calibration technique proposed in this paper. Thus, it is concluded that the proposed calibration technique sufficiently improves the navigation performance of RINS.

Finally, the calibration technique proposed in this paper performed system-level indirect calibration only for the *Z*–axis up position condition and the *Z*–axis down position condition to identify the gyro bias that changes according to the position of the IMU. Therefore, future studies will improve the navigation performance of RINS in a way that identifies and compensates for the gyro bias that changes according to the position of the IMU for more diverse positional conditions.

## Figures and Tables

**Figure 1 sensors-22-08355-f001:**
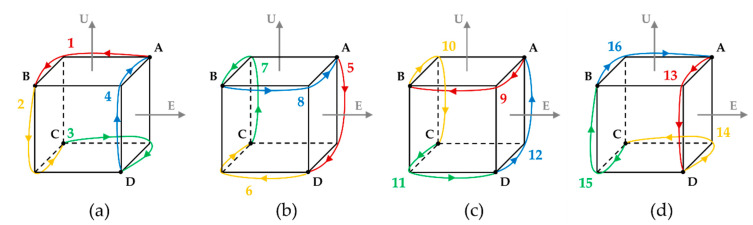
(**a**) Rotation sequence 1~4; (**b**) rotation sequence 5~8; (**c**) rotation sequence 9~12; (**d**) rotation sequence 13~16.

**Figure 2 sensors-22-08355-f002:**
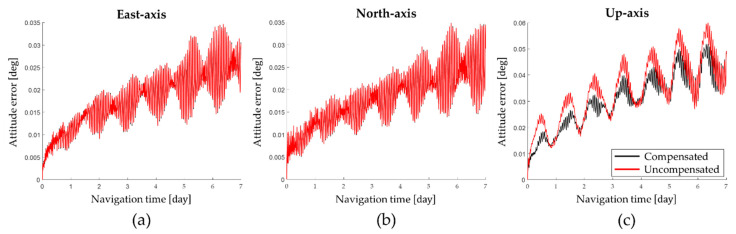
Attitude error according to the gyro bias error: (**a**) East−axis; (**b**) North−axis; (**c**) Up−axis.

**Figure 3 sensors-22-08355-f003:**
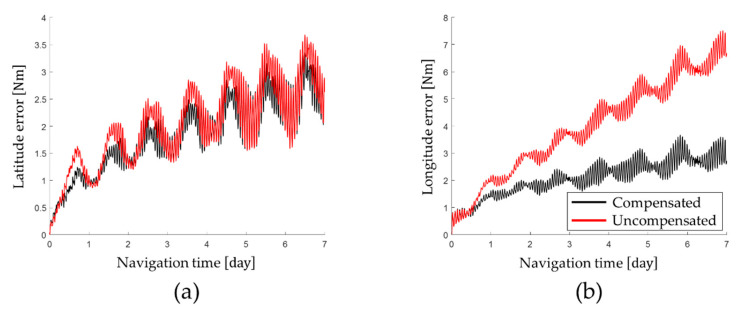
Position error according to the gyro bias error: (**a**) latitude error; (**b**) longitude error.

**Figure 4 sensors-22-08355-f004:**
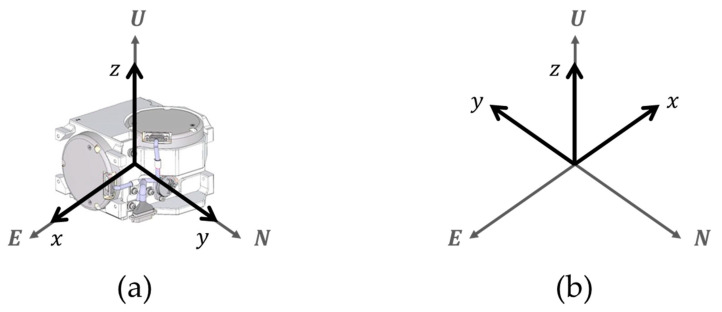
Rotation sequence in the *Z*−axis up position: (**a**) initial position; (**b**) position after rotation.

**Figure 5 sensors-22-08355-f005:**
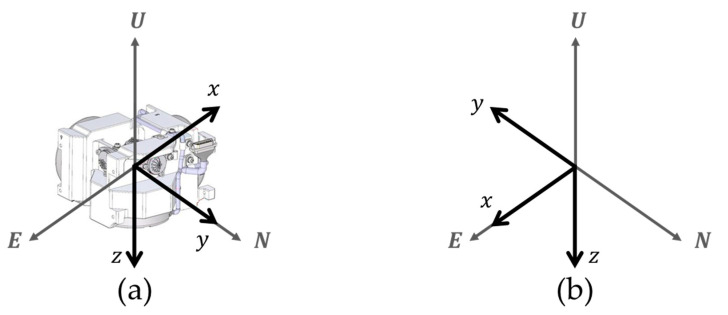
Rotation sequence in the *Z*−axis down position: (**a**) initial position; (**b**) position after rotation.

**Figure 6 sensors-22-08355-f006:**
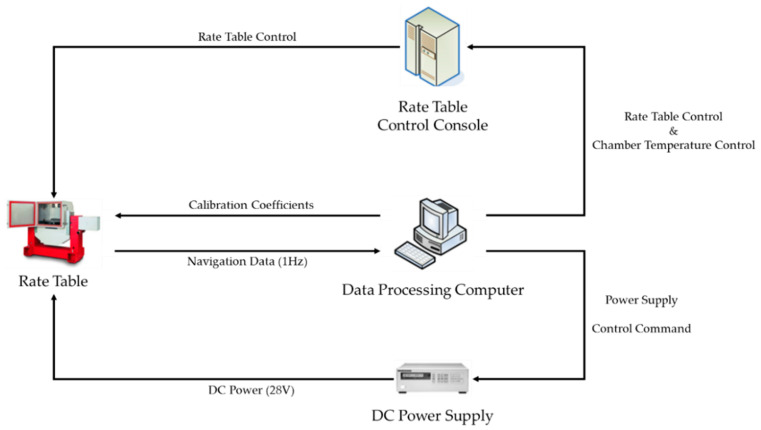
Calibration test diagram.

**Figure 7 sensors-22-08355-f007:**
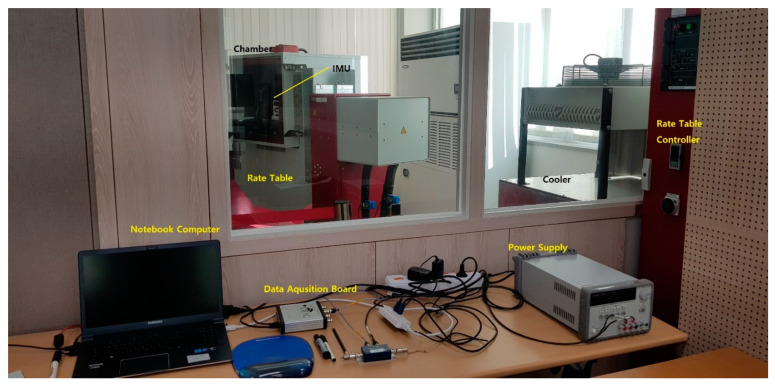
Experimental setup.

**Figure 8 sensors-22-08355-f008:**
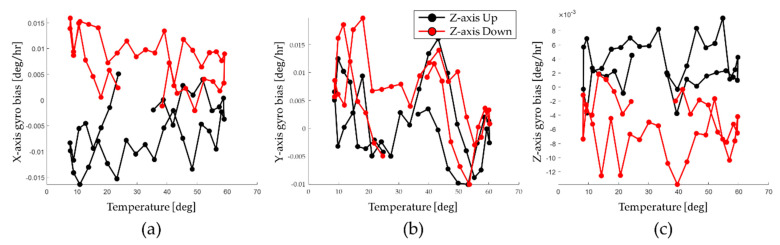
Identified gyro bias according to the position; (**a**) *X*−axis; (**b**) *Y*−axis; (**c**) *Z*−axis.

**Figure 9 sensors-22-08355-f009:**
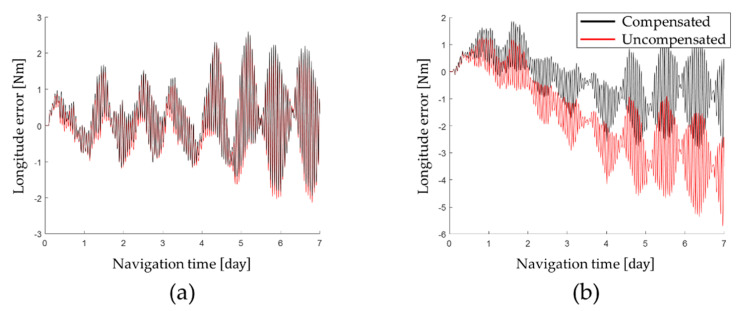
Position error after compensating for gyro bias transition: (**a**) latitude error; (**b**) longitude error.

**Figure 10 sensors-22-08355-f010:**
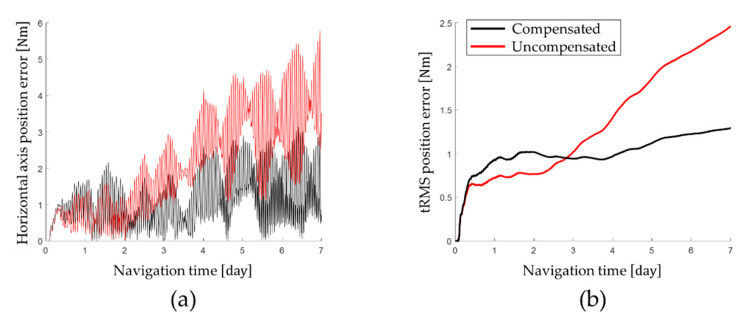
Position error after compensating for gyro bias transition: (**a**) latitude error and longitude error; (**b**) horizontal axis position error and tRMS position error.

**Table 1 sensors-22-08355-t001:** Table definition.

Symbol	Frames
b	The body frame
e	The earth frame
n	The navigation frame
i	The orthogonal inertial frame

**Table 2 sensors-22-08355-t002:** Direction cosine matrix for the 16-position rotation scheme.

${\left(C_{b}^{n}\right)}_{1}$	${\left(C_{b}^{n}\right)}_{2}$	${\left(C_{b}^{n}\right)}_{3}$	${\left(C_{b}^{n}\right)}_{4}$
$\left[\begin{array}{ccc} \cos\left(\omega_{r}t\right) & -\sin\left(\omega_{r}t\right) & 0\\ \sin\left(\omega_{r}t\right) & \cos\left(\omega_{r}t\right) & 0\\ 0 & 0 & 1 \end{array}\right]$	$\left[\begin{array}{ccc} -1 & 0 & 0\\ 0 & -\cos\left(\omega_{r}t\right) & \sin\left(\omega_{r}t\right)\\ 0 & \sin\left(\omega_{r}t\right) & \cos\left(\omega_{r}t\right) \end{array}\right]$	$\left[\begin{array}{ccc} -\cos\left(\omega t\right) & -\sin\left(\omega_{r}t\right) & 0\\ -\sin\left(\omega t\right) & \cos\left(\omega_{r}t\right) & 0\\ 0 & 0 & 1 \end{array}\right]$	$\left[\begin{array}{ccc} 1 & 0 & 0\\ 0 & -\cos\left(\omega_{r}t\right) & -\sin\left(\omega_{r}t\right)\\ 0 & \sin\left(\omega_{r}t\right) & -\cos\left(\omega t\right) \end{array}\right]$
${\left(C_{b}^{n}\right)}_{5}$	${\left(C_{b}^{n}\right)}_{6}$	${\left(C_{b}^{n}\right)}_{7}$	${\left(C_{b}^{n}\right)}_{8}$
$\left[\begin{array}{ccc} 1 & 0 & 0\\ 0 & \cos\left(\omega_{r}t\right) & \sin\left(\omega_{r}t\right)\\ 0 & -\sin\left(\omega_{r}t\right) & \cos\left(\omega_{r}t\right) \end{array}\right]$	$\left[\begin{array}{ccc} \cos\left(\omega_{r}t\right) & \sin\left(\omega_{r}t\right) & 0\\ \sin\left(\omega_{r}t\right) & -\cos\left(\omega_{r}t\right) & 0\\ 0 & 0 & 1 \end{array}\right]$	$\left[\begin{array}{ccc} -1 & 0 & 0\\ 0 & \cos\left(\omega_{r}t\right) & -\sin\left(\omega_{r}t\right)\\ 0 & -\sin\left(\omega_{r}t\right) & -\cos\left(\omega_{r}t\right) \end{array}\right]$	$\left[\begin{array}{ccc} -\cos\left(\omega_{r}t\right) & \sin\left(\omega_{r}t\right) & 0\\ -\sin\left(\omega_{r}t\right) & -\cos\left(\omega_{r}t\right) & 0\\ 0 & 0 & 1 \end{array}\right]$
${\left(C_{b}^{n}\right)}_{9}$	${\left(C_{b}^{n}\right)}_{10}$	${\left(C_{b}^{n}\right)}_{11}$	${\left(C_{b}^{n}\right)}_{12}$
$\left[\begin{array}{ccc} \cos\left(\omega_{r}t\right) & \sin\left(\omega_{r}t\right) & 0\\ -\sin\left(\omega_{r}t\right) & \cos\left(\omega_{r}t\right) & 0\\ 0 & 0 & 1 \end{array}\right]$	$\left[\begin{array}{ccc} -1 & 0 & 0\\ 0 & -\cos\left(\omega_{r}t\right) & -\sin\left(\omega_{r}t\right)\\ 0 & -\sin\left(\omega_{r}t\right) & \cos\left(\omega_{r}t\right) \end{array}\right]$	$\left[\begin{array}{ccc} -\cos\left(\omega_{r}t\right) & \sin\left(\omega_{r}t\right) & 0\\ \sin\left(\omega_{r}t\right) & \cos\left(\omega_{r}t\right) & 0\\ 0 & 0 & -1 \end{array}\right]$	$\left[\begin{array}{ccc} 1 & 0 & 0\\ 0 & -\cos\left(\omega_{r}t\right) & \sin\left(\omega_{r}t\right)\\ 0 & -\sin\left(\omega_{r}t\right) & -\cos\left(\omega_{r}t\right) \end{array}\right]$
${\left(C_{b}^{n}\right)}_{13}$	${\left(C_{b}^{n}\right)}_{14}$	${\left(C_{b}^{n}\right)}_{15}$	${\left(C_{b}^{n}\right)}_{16}$
$\left[\begin{array}{ccc} 1 & 0 & 0\\ 0 & \cos\left(\omega_{r}t\right) & -\sin\left(\omega t\right)\\ 0 & \sin\left(\omega_{r}t\right) & \cos\left(\omega t\right) \end{array}\right]$	$\left[\begin{array}{ccc} \cos\left(\omega_{r}t\right) & -\sin\left(\omega_{r}t\right) & 0\\ -\sin\left(\omega_{r}t\right) & -\cos\left(\omega_{r}t\right) & 0\\ 0 & 0 & -1 \end{array}\right]$	$\left[\begin{array}{ccc} -1 & 0 & 0\\ 0 & \cos\left(\omega_{r}t\right) & \sin\left(\omega_{r}t\right)\\ 0 & \sin\left(\omega_{r}t\right) & -\cos\left(\omega_{r}t\right) \end{array}\right]$	$\left[\begin{array}{ccc} -\cos\left(\omega_{r}t\right) & -\sin\left(\omega_{r}t\right) & 0\\ \sin\left(\omega_{r}t\right) & -\cos\left(\omega_{r}t\right) & 0\\ 0 & 0 & 1 \end{array}\right]$

**Table 3 sensors-22-08355-t003:** Sensor specification for simulation.

Parameter		Value
Gyro	Random Walk $\left[\deg/\sqrt{\mathrm{hr}}\right]$	0.003
	Bias [deg/hr]	0.01
	$\delta \beta_{x},\delta \beta_{y},\delta \beta_{z}$ [deg/hr]	0.005
	Scale Factor [ppm]	5
	Misalignment [arcsec]	5
Accelerometer	Bias [μg]	50
	Scale Factor [ppm]	50
	Misalignment [arcsec]	5

**Table 4 sensors-22-08355-t004:** The results of the navigation simulation.

	Uncompensated	Compensated
Latitude error	2.7088 Nm	2.6382 Nm
Longitude error	6.6784 Nm	2.6241 Nm

**Table 5 sensors-22-08355-t005:** Navigation performance of the long-term static state test.

	Uncompensated	Compensated
Latitude error	0.3241 Nm	0.3319 Nm
Longitude error	−3.5077 Nm	−0.6271 Nm
Horizontal-axis position error	3.5227 Nm	0.7096 Nm
tRMS position error	2.4603 Nm	1.2921 Nm

## Data Availability

Data are available upon reasonable request to the corresponding author.

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
