# Peer review of "Analysis of Gyro Bias Depending on the Position of Inertial Measurement Unit in Rotational Inertial Navigation Systems"

_sensors, 2022, doi:10.3390/s22218355_

Round 1
Reviewer 1 Report
please check the attached file.

Author Response
Dear Reviewer,
Thank you for giving us the opportunity to submit a revised draft of the manuscript “Analysis of Gyro Bias Depending on the Position of Inertial Measurement Unit in Rotational Inertial Navigation System” for publication in the Sensors. We appreciate the time and effort that the editor and the reviewers dedicated to providing valuable feedbacks and insightful comments on our manuscript. The comments are all encouraging and very helpful for revising and improving our manuscript. We have been able to incorporate changes to reflect most of the suggestions provided by the reviewers. We hope the revised manuscript will better suit your high standards. The authors welcome further constructive comments if any.
The corrections in the manuscripts to the reviewer’s comments are provided below. All page numbers refer to the revised manuscript file with tracked changes.
[Comment 1] This work focus on gyro bias error suppression of RINS. Advise to reduce the corresponding description on acc or adjust the title as IMU analysis.
Response: The title of the previously submitted manuscript is focused on the gyro bias. Therefore, in this paper, the authors focused on the gyro bias depending on the position of IMU. However, in addition to gyro bias depending on the position of IMU, (such as gyro bias, scale factor, misalignment and accelerometer bias, scale factor, and misalignment error) the calibration coefficients affect the navigation performance of RINS. In this paper, the authors mathematically described that calibration coefficients, except gyro bias depending on the position of IMU, are compensated by the dual-axis 16 position rotation scheme.
To avoid confusion to the reader, the authors reinforced the description of the compensated calibration coefficients through the dual-axis 16 position rotation scheme, and emphasized more on the gyro bias depending on the position of IMU. Therefore, the authors think that the title of this manuscript can properly express the study.
The corresponding description on accelerometer is detailed on line 223 to 227, line 258 to 263 and line 315 to 317.
[Comment 2] According to (3) and (9), inertial device error, especially gyro, is affected by kinds of environment factor. These bias, scale factor errors can be established as function of environment disturbances, which is regarded as essential description of these error. Advise to add another corresponding error compensation method ref. to make reading convincing.
Response: In order to make more convenience reading, the authors adopted the error compensation methods that recommended by reviewer.
The navigation performance of the RINS is most affected by bias and random walk. In the case of scale factor, it has a relatively small effect compared to bias. In addition, random walk has characteristics that cannot be compensated. Therefore, this paper improved the navigation performance of the RINS by precisely compensating bias and damping scale factor.
In addition, the authors agree that there is a lack of literature reviews about error compensation methods. In the revised manuscript, contents of the error compensation method references to make reading convincing are improved.
The renewed references are included on line 34 to 62 and line 517 to 518, 521 to 522, 569 to 570.
The authors sincerely hope that the improved descriptions can be accepted for publication in the Sensors.
[Comment 3] Gyro bias error identification is induced in (17)-(19). Please add the corresponding experimental verification.
Response: In the previously submitted manuscript, the description on verifications for the proposed calibration method have not been discussed. The authors reinforced the descriptions of the relationship between gyro bias identification and verification test, and improved the manuscript for readers.
In the revised manuscript, gyro bias depending on the position of IMU could be calculated through Equations (17) to (19). Proposed calibration method compensates the gyro bias depending on the position of IMU according to the Z-axis Up and Z-axis Down conditions during the operation phase of RINS. The performance of the proposed calibration method is verified by performing stationary test in Chapter 4.
In the revised manuscript, gyro bias identification and corresponding experimental verification are detailed on line 362 to 375 and line 401 to 405.
Reviewer 2 Report
This paper introduces a new compensation method for the gyro bias affected by changes in temperature distribution, direction of gravity, and dithering according to the rotation of IMU. Simulation and test are both conducted, and the results obtained suggest the effectiveness of the proposed method in improving the navigation performance of RINS. However, some comments need to be well addressed before acceptance:
1. In the paper, numerical simulation is a major approach to predict the calibration effect, but not any detailed method or procedure is presented. This may lead to unconvincing results. Please add some description.
2. In the stationary state test, how was the compensation realized? Please give an illustration.
3. A real-taken photograph of the experiment set-up should be provided to display the test process.
4. What’s the meaning of discussing different axis frames in figures 4 and 5?
5. The literature review is not comprehensive enough. Some critical research, especially on the advanced vibratory ring gyroscopes, is not concerned. The authors should make a particular comment on this aspect based on following works: International Journal of Mechanical Sciences, 2020, 187: 105915; Acta Mechanica Solida Sinica, 2021, 34(1): 65-78.
6. The paper has some writing issues, e.g., “compensate” in line 16 should be “compensates”; “Equations 13 and 15” in line 338 should be “Equations (13) and (15)”. The authors should carry out a thorough check on the paper.
Author Response
Dear Reviewer,
Thank you for giving us the opportunity to submit a revised draft of the manuscript “Analysis of Gyro Bias Depending on the Position of Inertial Measurement Unit in Rotational Inertial Navigation System” for publication in the Sensors. We appreciate the time and effort that the editor and the reviewers dedicated to providing valuable feedbacks and insightful comments on our manuscript. The comments are all encouraging and very helpful for revising and improving our manuscript. We have been able to incorporate changes to reflect most of the suggestions provided by the reviewers. We hope the revised manuscript will better suit your high standards. The authors welcome further constructive comments if any.
The corrections in the manuscripts to the reviewer’s comments are provided below. All page numbers refer to the revised manuscript file with tracked changes.
[Comment 1] In the paper, numerical simulation is a major approach to predict the calibration effect, but not any detailed method or procedure is presented. This may lead to unconvincing results. Please add some description.
Response: As the reviewer pointed out, numerical simulation is critical approach to predict the calibration effect. However, the submitted manuscript have not been discussed the procedure of simulation. In the revised manuscript, the simulation procedure is more detailed to lead convincing results.
In this paper, the simulation is performed to evaluate the effect of gyro bias depending on the position of IMU on navigation performance of RINS. The results of numerical simulation showed that RINS based on dual-axis 16 position rotation scheme could not compensate the Z-axis gyro bias depending on the position of IMU. Based on the simulation results, proposed calibration method designed the calibration test for Z-axis up and Z-axis down conditions.
The numerical simulations to predict the calibration effect are described on line 261 to 263 and line 396 to 405.
The authors sincerely hope that the novelty can be accepted for publication in the Sensors.
[Comment 2] In the stationary state test, how was the compensation realized? Please give an illustration.
Response: The stationary state test is performed to evaluate the improved navigation performance of the proposed calibration method. The stationary state test is designed to performing navigation for about 7 days on the leveled surface. The navigation process is repeated for 2 conditions – applying proposed calibration method and applying conventional calibration method. During the stationary state test, the gyro bias depending on the position of IMU calculated in Section 4.1 is compensated for real-time.
For real-time compensation, the calculated gyro bias sets are approximated as third-order polynomial function for temperature. Since position of the IMU could be estimated during the operation phase of RINS, we can compensate the gyro bias depending on the position of the IMU for Z-axis Up and Z-axis Down.
The authors went through the entire manuscript to improve the descriptions of stationary state test and evaluation simulations. We sincerely hope that the improved descriptions can be accepted for publication in the Sensors.
The realization of the compensation in the stationary state test is described on line 437 to 441.
[Comment 3] A real-taken photograph of the experiment set-up should be provided to display the test process.
Response: In the previously submitted manuscript, the experiment set-up is described based on diagram. To describe the detailed experiment set-up, the authors supplied a real-taken photograph in the revised manuscript.
The supplied photograph and improved experiment set-up are detailed on line 384 to 395.
[Comment 4] What’s the meaning of discussing different axis frames in figures 4 and 5?
Response: The authors revised the manuscript to improve the discussions in figure 4 and 5. To analyze the rotation sequence, Figure 4 and 5 applies some frames such as body frame and navigation frame. The body frame is applied to describe the position, velocity, and orientation of the sensor platform. The body frame consists of an origin that is typically placed at the center of gravity and three orthogonal axes. On the other hand, the navigation frame is a navigation frame that fixed to the platform and moves with the body frame. In the revised manuscript, these frames are described in the first column of Section 2.
The meaning of discussing different axis frames in Figure 4 and 5 are detailed on line 134 to 147 and line 336 to 338.
[Comment 5] The literature review is not comprehensive enough. Some critical research, especially on the advanced vibratory ring gyroscopes, is not concerned. The authors should make a particular comment on this aspect based on following works: International Journal of Mechanical Sciences, 2020, 187: 105915; Acta Mechanica Solida Sinica, 2021, 34(1): 65-78.
Response: This paper proposed a calibration method to improve the navigation performance of RLG-based RINS. The authors agree that there is a lack of literature reviews that precede the study. In the revised manuscript, contents of the advanced vibratory ring gyroscopes are improved.
The improved literature reviews are described on line 00 to 00 and line 61 to 62.
The authors sincerely hope that the improved literature reviews can be accepted for publication in the Sensors.
[Comment 6] The paper has some writing issues, e.g., “compensate” in line 16 should be “compensates”; “Equations 13 and 15” in line 338 should be “Equations (13) and (15)”. The authors should carry out a thorough check on the paper.
Response: The authors agree with the previously submitted manuscript has some typos. In the revised manuscript, the noted typos are corrected. In addition, the authors went through the entire manuscript to eliminate the typos.
The corrected typo is written on line 16 and line 343.
We hope the revised manuscript will better suit your high standards.
Round 2
Reviewer 2 Report
The paper has been improved and can be considered for publication.